# An Immersive Journey through Flawed Technology: Users' Perceptions of VR in Journalism

**Alexander Godulla, Rosanna Planer \*, Cornelia Wolf, Annika Lück and Fiona Vaaßen**

Institute for Communication and Media Studies, Leipzig University, 04109 Leipzig, Germany; alexander.godulla@uni-leipzig.de (A.G.); cornelia.wolf@uni-leipzig.de (C.W.); AnnikaLueck@gmx.net (A.L.); fiona.vaassen@gmx.de (F.V.)
\* Correspondence: rosanna.planer@uni-leipzig.de

**Abstract:** Virtual reality (VR) has had the reputation of being a revolutionising technology ever since it emerged in the early 1960s, but virtual is not yet a successful reality in journalistic practice. Examining VR's current situation and the factors preventing it from reaching its predicted potential in digital journalism, this paper analyses the user comments ($n = 770$) on 15 journalistic VR apps offered by media organizations, with the help of a qualitative-reductive content analysis. Deductive categories of analysis contain the constructs of immersion, emotion, usability, and utility, which are further specified by inductive subcategories in the course of the analysis. Results show that users positively highlight different aspects of emotion and immersion that the VR apps elicit, and criticize journalistic VR apps for their low levels of utility and usability. Implications for journalistic practice and research are subsequently drawn.

**Keywords:** virtual reality; digital journalism; VR app; user comments; emotion; immersion; usability

## 1. Introduction

For decades, VR researchers have been fascinated by VR technology as a possibility to explore the space between reality and virtuality. VR is defined as "the use of computer technology to create the effect of an interactive three-dimensional world in which the objects have a sense of spatial presence" (Bryson 2013, p. 1). It typically uses headsets to generate realistic images, sounds, and sensations that simulate a user's physical presence in a virtual environment (Wimmer 2017), usually combined with two touch controllers that allow the user to move within the virtual space. Despite its long-held promise of creating an alternative reality, annoying shortcomings in VR technology remain to this day: "[T]he bulkiness and grainy screens of current-generation headsets, the annoyance of getting a cord wrapped around your ankles, the likelihood that you'll accidentally ram your hand into some furniture, or the frustration of setting up new and sometimes complicated hardware" (Robertson 2020) make a breakthrough of VR technology still questionable.

This directly affects VR's use in immersive journalism (Baía Reis and Coelho 2018), which is "an experimental approach that allows users to experience, and subsequently be immersed in, stories created not in the real world but in a virtual, augmented, or mixed reality" (Gynnild et al. 2021, p. 2), and where initial expectations were high. Back in 2015, TIME magazine put VR on its cover, referring to the "surprising joy of virtual reality" (Time 2015). Back then, a huge market was predicted for VR, and still is today (Frausto-Robledo 2018). Facebook has believed in this prediction for some years now, having bought the company Oculus VR for USD 2 billion in 2014. Since then, through continuous research and an aggressive pricing policy, Facebook has succeeded in making Oculus headsets the most widely used VR glasses on the popular platform Steam (Machkowech 2020). Each of these headsets automatically has access to the in-house Oculus Store, which furthermore attracts a wide audience. Therefore, Oculus is often referred to as the most promising approach to mass-market VR (Pavlik 2015). Other journalistic media outlets—such as ARTE, the

BBC, the Swiss newspaper *Blick*, the Argentinian newspaper *Clarín*, the Italian newspaper *Corriere della Sera*, the British publication *The Economist*, or the German public television broadcaster ZDF (Ambrosio and Fidalgo 2019, p. 6f)—became increasingly engaged in the production of journalistic VR projects. In late 2019, however, one driver of journalistic VR productions—the BBC—disbanded the team it had initially hired to create VR content in its BBC VR Hub, saying that its funding had ended (BBC 2019).

For digital journalism as "the networked production, distribution, and consumption of news and information" (Waisbord 2019, p. 351), VR seems to be a risky business. Although the technology does not lack creativity, money, or vision, and the list of potentials and qualities—especially in the journalistic field—seems long and promising, the technology's history is shaped by failed attempts. The question of whether a turning point is yet to occur remains; a breakthrough in technology could be one key to generating revenues in digital journalism.

While "VR is the result of a set of technological advances and long decades of research that have resulted in a wide range of works that try to address this phenomenon from very diverse areas of knowledge" (Ambrosio and Fidalgo 2019, p. 5), the question of which aspects of VR currently hold the technology back from becoming a more widely used journalistic asset still remains unanswered, and is thus at the centre of the research project underlying this paper. The focus therefore lies on news users and their perception of VR applications in journalism, since "this type of innovation is about what audiences (allegedly) need and want, so the audience needs to tell us what to make of these innovations" (Lecheler 2020, p. 289). Taking this angle, an extensive qualitative analysis of the user comments ($n = 770$) on VR applications gives insights into the strengths and weaknesses of VR in journalism as evaluated by its users, as well as possible implications for future productions. In order to comprehensively understand the present role of VR in digital journalism—and possibly to shape its future—we must shed light on the historical development of the technology, the underlying theoretical concepts, and the specific qualities and potentials shaping its journalistic discourse.

## 2. State of Research

### 2.1. The Visionary Origins of VR Technology

The roots of VR go back to 1962, when Morton Heilig, an American cinematographer and the "Father of Virtual Reality" (USC 2020), introduced a—sadly, unsuccessful—simulator called "Sensorama", an "apparatus to stimulate the senses of an individual to stimulate an actual experience realistically" (Heilig 1962, p. 1). Although Heilig's Sensorama did not succeed back then—not least because of its eccentric construction—it fuelled further innovations in the field of VR, such as the head-mounted display (HMD), introduced in 1965 by Ivan Sutherland. The HMD allowed for an immersive experience within a computer-generated 3D world, and resembled contemporary headsets, with the main difference being that the apparatus was so heavy that it had to be attached to the ceiling (Schart and Tschanz 2015, p. 26).

Throughout VR's history, the question of how real we perceive the virtual to be is one of the most important denominators, eventually—and in theory—resulting in a "suspension of disbelief" (Hand 1994): "We must temporarily remove any doubt about the reality of the experience ( . . . ) in order that the user may interact as though the experience were real". Today, this fact seems to account for "one of the most remarkable aspects of immersive virtual environments", which is that "people tend to respond realistically to virtual situations and events even though they know that these are not real" (De la Peña et al. 2010, p. 293). For instance, users of VR applications objectively have no reason to experience motion sickness, since they are standing safely in a room, wearing VR glasses—but they often report feeling nausea or motion sickness. When embedded into a non-fiction narrative, this response opens up a wide range of possibilities for distributing and conveying important journalistic content, having users experience and understand a situation better through personal experience.

The latter is referred to as immersion, a concept that early on was defined by Morat (1998) as one of the key principles of VR, referring to the complete submersion of the user into an artificial environment, while the interface—the intersection of the person and the "reality machine"—dissolves into the virtual (Morat 1998, p. 38). It can be referred to as "an objective measure of the extent to which the system presents a vivid virtual environment while shutting out physical reality" (Cummings and Bailenson 2016, p. 274). Since the user is not only immersed in this world passively, but can move around and take part in it, the principles of interaction and navigation seem crucial (Morat 1998, p. 38). Morat's (1998) forecasting of future VR technology was no less imaginative than Heilig's Sensorama: Morat predicted human beings experiencing VR while being embedded into whole-body suits (ibid). This rather fanciful prediction has not proved true, but the concept of immersion still plays a relevant role in scholarly discourse about VR.

### 2.2. Immersion in Digital Journalism: The Feeling of 'Being There'

"The combination of VR and journalism has led to the emergence of so-called 'immersive journalism'" (Ambrosio and Fidalgo 2019, p. 6), which is defined as "the production of news in a form in which people can gain first-person experiences of the events or situations described in news stories" (De la Peña et al. 2010, p. 291). It thereby "immerses users into another reality (the news story), takes over their attention and makes them feel part of it, which leads them to react within that virtual environment as they would do in the physical world" (Ambrosio and Fidalgo 2019, p. 6). "The fundamental idea of immersive nonfiction is to allow the audience to actually enter a virtually recreated scenario representing the story" (De la Peña 2011, p. 1).

While Hand (1994) introduced the aforementioned concept of immersion as a "suspension of disbelief", which might lead to the critical assumption that users have to give up their agency or control when they enter a virtual environment, Baía Reis and Coelho (2018) state that immersion is "not the prim suspension of disbelief, but its joyous capsizing" (Baía Reis and Coelho 2018, p. 1090). Hence, today, there is a focus on the cognition and agency of the user: "Rather than immersion being an external factor given to users, immersion is a fluid state that is processed and determined by users" (Shin and Biocca 2017, p. 2812). With that notion comes an inherent subjectivity that makes immersion hard to grasp and to study: "I cannot experience your immersion, and you cannot experience mine" (Hassan 2020, p. 207).

There are more technology-centred approaches to immersion, however, such as that introduced by Cummings and Bailenson (2016). The authors distinguish between immersion as an objective, technological quality of media and, in contrast, the concept of presence as "a psychological experience of being there" (Cummings and Bailenson 2016). On the other hand, De la Peña et al. (2010, p. 294) view immersion as "the sense of being in the place depicted by the virtual displays". Interestingly, the debate as to whether immersion is more a technology-driven or user-driven concept is not as new as it might seem. As far back as 1994 there was a fear of "placing too strong an emphasis on the technology, as this may only lead us into a dead end" (Hand 1994). Taking both approaches together, immersion is a concept provided by technology—such as VR features—and experienced by the user in terms of a feeling of presence. The concept pf presence can be regarded as a "two-dimensional construct" (Cummings and Bailenson 2016, p. 274), where "the media users build a mental representation of the space portrayed by the media product ( … ) [and] position themselves and realized action possibilities within that space" (Wirth et al. 2007, p. 517). As Biocca and Levy put it in 1995, "much of immersion is related to user subjectivity and the objectivity of the technology" (Biocca and Levy 1995); hence, both aspects have to be taken into account when producing journalistic VR content, and when studying VR in journalism.

The immersive technology renders agency to the user in such a way that it becomes a "defining characteristic of VR, which gives users both real and perceived control over, and responsibility for, actions to which the system then responds" (Mabrook and Singer

2019, p. 2098). At the end of that equation, Van Damme et al. (2019) postulate that "the more a device captivates senses and blocks out stimuli from the physical world, the more it is considered immersive (Van Damme et al. 2019, p. 2056). Again, this was already being stated back in 1998, when Morat found that the more perfect the illusion, the more the user would be excluded from the outer world (Morat 1998, p. 40). Focusing on the subjectivity of the user's experience—the feeling of being there, the senses, and feeling part of a news story—it becomes obvious that VR is closely connected to feelings, emotion, and empathy.

### 2.3. The Emotional 'Empathy Machine' and Its Requirements

Since Chris Milk's TED Talk entitled "How virtual reality can create the ultimate empathy machine" (Milk 2015), both media practitioners and scholars have picked up on the term "empathy machine" and investigated the degree to which VR might actually enhance the emotion of empathy (Alsever 2015; Hassan 2020; Lecheler 2020; Sánchez Laws 2020); however, the results seem divergent. Mabrook and Singer (2019) pick up on Milk's statement that VR might enhance empathy simply because users feel as if the occurrences are happening directly to them, and Sánchez Laws (2020) concludes that "it seems that some strands of immersive journalism are beginning to meet the requirements which enable us to witness the emotions of others and to thereby feel empathy for them" (p. 223). This opens up new pathways for digital journalism to enhance societal discourse by facilitating the understanding of a viewpoint that might be different from one's own. Furthermore, Shin and Biocca (2017) point out in their study that the notion of presence, as introduced earlier, can enhance empathy, which is confirmed by Van Damme et al. (2019), who show that the more the recipients feel present in a virtual environment, the more they feel empathy with the acting characters. On the other hand, however, Hassan (2020) states that "human agents are analogue agents from an analogue world" (p. 195), and argues that empathy "cannot be generated from a digital source" (Hassan 2020, p. 195). He concludes by saying that VR "has no place from which to emerge into consciousness, because the analogue self within an analogue world has been banished pro tem to the border of the closed system that immerses the visual and aural senses" (Hassan 2020, p. 209). Roose (2020) also doubts that VR can generate real-world empathy, stating that "these experiences aren't fully immersive".

While empathy may be the most outstanding and most frequently discussed emotion in this context, the bandwidth of emotions elicited by VR stretches much further, which is why a focus on emotion in general seems relevant. Lecheler (2020), too, observed this emotional turn in journalism, focusing on the connection between emotion-driven journalism and technological innovation, and pointing out, in relation to the limited audience studies conducted thus far, that "it is almost absurd not to systematically study the audience when looking at emotion-driven innovation in journalism" (Lecheler 2020, p. 289). One such study was carried out by Nielsen and Sheets (2019); following a uses and gratifications approach within a focus group study, they found that six different gratifications can be obtained from VR: immersion and transportation (subsumed as experience), emotion and empathy (subsumed as affect), and information and control (subsumed as agency) (Nielsen and Sheets 2019, p. 7). Users had reservations about the technology, but saw great potential for its use (Nielsen and Sheets 2019, p. 1). Meanwhile, an experimental study conducted by Kang et al. (2019) showed that groups using VR headsets and 360° videos in order to consume information rated content credibility higher than the control group in the study (p. 306), suggesting that VR technology offers the additional opportunity for digital journalism to counteract a potential lack of credibility.

Apart from aspects of technology and emotion, Sánchez Laws (2020) highlights that users should be provided with enough background information to be able to better interpret the presented situation. Palmer (2020), studying immersive 360° articles of *The New York Times*, confirms these findings, stating that users must receive sufficient background information to interpret and question the events depicted in order to be able to put themselves in the perspective of the persons involved.

These studies provide important insights into the audience perceptions of VR technology, and show that immersion, as a feeling of "being there", can be enhanced by notions of felt empathy, although questions remain as to whether genuine empathy can be derived from digital sources; these notions of empathy and, more generally speaking, emotion, again rely on or are intertwined with the technology, as well as with given background information. Since "the audience will necessarily have to be the message in immersive journalism" (Baía Reis and Coelho 2018), further audience studies need to be conducted in order to obtain a fuller picture of how audiences perceive immersive VR journalism today, with regards to the aforementioned emotion-, technology-, and content-related aspects.

### 2.4. VR Technology: Advanced, but Still Obsolete?

Back in the 1960s, VR technology was not yet advanced enough (Hand 1994) to deliver a breakthrough. Today's immersive devices are "miles ahead of their predecessors" (Roose 2020), increasingly cheap, and faster to produce—which, according to Hardee (2016), accelerates their incorporation into the communications sector. These devices can be VR applications and features, 360° videos, or animated 3D models (Mabrook and Singer 2019; Shin and Biocca 2017). Examples or expressions of immersive features might range from stereoscopic vision, tracking level, sound quality, and image quality to field of view and user perspective, as identified by Cummings and Bailenson (2016) through a meta-analysis.

Despite these developments, however, the current state of VR technology still does not seem to contain enough added value compared to other, already existing technologies: "The user confronts not a powerful immersion that leaves meaningful and lasting experience traces, nor an interactivity sphere where subject and object mutually transform each other, but digital spectacle" (Hassan 2020, p. 209). Even the leading systems still lack some basic features and, outside of gaming, "there isn't much you can do on a V.R. headset that you can't do more easily on another device" (Roose 2020). Scholars furthermore argue that most of what we refer to as VR are actually only 360° videos (Mabrook and Singer 2019; Nielsen and Sheets 2019). Hence, there are still major deficits in the usage of VR, which also make it hard for digital journalism to effectively make use of the technology.

In order to achieve the best possible usability, a user has to understand intuitively how to navigate through a digital offer (Meier 2003, p. 259), which must be visually appealing and well arranged (Godulla and Wolf 2017, p. 65). Not every story is suitable to be displayed with VR technology; for example, complex, nuanced stories (Mabrook 2021, p. 220) or "facts and statistics" (Nielsen and Sheets 2019, p. 9) are rather less suitable. This bears challenges for journalistic practice, where journalists have to apply the "media-content match" (Planer and Godulla 2020, p. 13). As promising as the concept of usability sounds in theory, it might be challenging in practice, and the same applies to the utility (Meier 2003) of the digital offer in terms of a present utilization value (Meier 2003, p. 259). If the VR story or VR application does not work out properly, utility is not achieved (Godulla and Wolf 2017, p. 65), begging the question: what is it good for? Roose (2020) also points to another challenge VR faces in terms of utility, stating that "much of what a V.R. headset offers can be found in other places", and arguing that humans are creatures of habit who might "simply prefer virtual experiences that don't require them to strap an expensive computer to their forehead" (Roose 2020).

However, "since approximately 2012, a new ecosystem of immersive virtual reality technologies and experiments has emerged" (Baía Reis and Coelho 2018, p. 1090), such as VR apps produced in the journalistic media field, which—especially against the background of failing attempts in VR technology—now have to be observed through an analytical lens. The introduced characteristics and potentials of VR that have developed over time appear highly interconnected and interdependent. Immersion, as one of the main arguments for VR and seemingly the main goal of VR production, depends on both the successful utility and usability of the technology, which again shapes the users' potential to feel empathy during the reception of a VR story. Integrating these concepts and picking up on the

importance, highlighted earlier, of combining the subjectivity of the user and the objectivity of the technology, this study aims to find answers to the following research question:

> How do users evaluate modern VR apps produced by journalistic media outlets, with regards to the qualities of immersion, emotion, usability, and utility?

## 3. Methodology

Scholars recently found that only "a small but growing number of productions using computer-generated virtual worlds designed to represent reality do exist" (Nielsen and Sheets 2019, p. 3), stating that these are "generally inaccessible or limited to eye control" (ibid.). Based on this, the present paper takes a novel approach in examining VR technology in terms of apps—which are both accessible and not limited solely to eye control; they can be referred to as "a gameful approach to immersive journalism" (Arjoranta et al. 2021, p. 142). On 24 April 2020, all VR apps available for the Oculus Rift S—the flagship product of Oculus—were assessed through the Oculus Store ($n$ = 1757). The next step was to see whether they had been published by a journalistic company, since the products of organizations primarily active as video game companies or movie studios were not addressed by the underlying research interest. After applying this criterion, a total of 15 VR apps produced by journalistic media outlets remained in the sample. They were regarded as comparable since they were not only produced by journalistic organizations, but also all circled around socially, technologically, culturally, historically, politically, or otherwise relevant topics. The majority was provided by the British public service broadcaster the BBC, which alone offered eight apps. Table 1 provides an overview of the distribution of titles by provider and various parameters, such as the content, the number of comments, and the publication date. Most of the apps were published in 2017, with the most recent one being published in 2019. The fact that these apps have existed at least for some years by now, and are frequently used and still on the market, implies that the media organizations did not actively decide to take them down, which justifies their analysis in this project; they mark a point in time in which journalistic media organizations experimented with VR apps.

**Table 1.** Sample of VR apps produced by journalistic media outlets.

| Publisher | Nr. | Title | | Publication Date | Comments |
|---|---|---|---|---|---|
| ARTE | 1 | Notes on Blindness | Society: A journey through the world of a blind person. | 7 December 2017 | 19 |
| BBC | 2 | 1943 Berlin Blitz | History: World War II: Inside a Lancaster bomber flying over Berlin in 1943. | 4 October 2018 | 215 |
| | 3 | BBC Home—A VR Spacewalk | Technology: A spacewalk inspired by NASA's training program. | 30 November 2017 | 159 |
| | 4 | Bear Island | Environment: A black bear searching for a fishing spot. | 29 March 2017 | 62 |
| | 5 | Cat Flight | Environment: A mother caracal trying to find food in a desert. | 23 February 2017 | 21 |
| | 6 | Easter Rising | History: Memories from 1916 Easter Rising in Irish History. | 5 July 2017 | 15 |
| | 7 | Is Anna OK? | Society: An accident that occurred to 20-year-old twins. | 18 October 2018 | 18 |
| | 8 | Oogie | Environment: The journey of a small beetle through the South African desert. | 5 April 2017 | 26 |
| | 9 | We wait | Politics: Experiences of migrants during refugee crisis. | 13 December 2016 | 49 |

**Table 1.** *Cont.*

| Publisher | Nr. | Title | | Publication Date | Comments |
|---|---|---|---|---|---|
| CNN | 10 | CNN VR | Technology: Inside a newsroom of the future. | 15 March 2018 | 46 |
| LIVE | 11 | Buzz Aldrin: Cycling Pathways to Mars | Technology: American astronaut Aldrin's ideas about inhabiting planet Mars | 16 June 2017 | 30 |
| Sky Ltd. | 12 | Sky VR: Hold the World | Culture: A journey through London's Natural History Museum with Sir David Attenborough. | 1 November 2018 | 39 |
| | 13 | Sky VR | Miscellaneous: Latest Sky VR content. | 3 October 2016 | 22 |
| NYT | 14 | Apollo 11: As They Shot It | History: A journey to the moon in the footsteps of Neil Armstrong and Buzz Aldrin. | 3 December 2019 | 43 |
| VICE | 15 | Cut-Off | History: Prime Minister Justin Trudeau's historic visit to Shoal Lake 40 | 26 October 2016 | 6 |

All user comments on the 15 identified apps and available at the time of the analysis were saved from the Oculus Store ($n = 770$). These comments ranged from merely one word in length to lengths of several sentences, and were fully incorporated into a qualitative-reductive structuring content analysis, following the methodology of Mayring (2015). Since analysing social media data and user comments in research poses important ethical concerns (Fossheim and Ingierd 2015, p. 9), preserving the anonymity of the evaluated comments was guaranteed. The comments were posted publicly without direct traceability to the users; hence, they were evaluated as belonging to the public sphere, as opposed to private comments (McKee and Porter 2009, p. 88). Furthermore, topic sensitivity, the degree of interaction, and the subject vulnerability were low (McKee and Porter 2009, p. 88), which is why consent was not deemed essential. Furthermore, "from a liberal perspective, the comments are a sign of the political times and the deep globalization of sources of information" (Castellano Parra et al. 2020, p. 87).

Based on the introduced concepts that journalistic research has identified as crucial characteristics of VR and immersive journalism, the research material was approached through a qualitative coding scheme with the four deductive categories of immersion (I), emotion (II), usability (III), and utility (IV). Two coders coded the data using the qualitative data analysis software MAXQDA. Before starting the coding process, they received coding training. In the first round of coding, all mentions of one of the four deductive categories were marked. During this process, another major inductive category—discourse (V; marked with a * below)—was added, since discourse-related comments appeared frequently and, thus, appeared relevant, especially against the background of journalism's function to serve as a platform for critical discourse. In a second round of coding, the coders derived subcategories for each deductive category from the coded material. Table 2 shows an overview of the final deductive and inductive categories. In a third step, all comments were individually checked for the presence of a positive or negative tone.

**Table 2.** Overview of deductive and inductive categories of analysis.

| Deductive | Inductive |
|---|---|
| | (1) Immersive character of the app/story |
| | (2) Perception of audio-visual effects/sensuality |
| **I Immersion** | (3) Storytelling or script that causes immersion |
| | (4) Background information that causes immersion |
| | (5) Technical implementation that causes immersion |
| | (5) Emotional character of the app/story |
| **II Emotion** | (6) Perception of audio-visual effects/sensuality |
| | (7) Storytelling or script that causes emotion |
| | (8) Background information that cause emotion |
| | (9) Perception of audio-visual effects/sensuality |
| | (10) Motion sickness |
| **III Usability** | (11) Navigation/controller/mode |
| | (12) Interactivity |
| | (13) Background information |
| | (14) Language |
| | (15) (Technological) requirements for installation |
| **IV Utility** | (16) Technical factors |
| | (17) Technical implementation of audio-visual effects (sensuality) |
| **V Discourse \*** | (18) Political discourse/propaganda |
| | (19) Fake news |

\* Main category inductively derived while coding.

In the following, the results of the main categories will be introduced with the help of the generated inductive categories. These categories are considered to be a key result of this study, from which further quantitative studies can depart, since they represent what VR app users find worth mentioning. These inductive categories are focused on and explained through examples of user comments, rather than focusing on the individual apps and their numbers. This method highlights the existing trends and evaluations across the apps, since it is not an evaluation of the individual apps that is of interest, but the overall evaluation of the five main categories across the whole field.

## 4. Findings

### 4.1. Immersion: "It Really Feels Like Being There"

For the main category immersion, five inductive subcategories emerged from structuring the user comments: The first was the general immersive character of the app (1), where users described the VR experiences as "extremely immersive" (Home) and "truly immersive" (1943 Berlin Blitz), with a surprisingly immersive "sensation of weightlessness" (Home). The second was the perception of audio-visual effects (2); here, users highlighted the "authentic radio broadcast" (1943 Berlin Blitz), and how the experience "does bring the audio to life" (1943 Berlin Blitz). The video content was also evaluated positively overall, with one user stating that "this is the first immersive video that I've seen where the video quality was high enough that I wasn't taken out of the experience [ . . . ]" (Cut-Off). In one app, "David Attenborough was recorded in 360 degrees with 160 cameras to place his recording inside virtual reality and it is uncanny" (Sky VR). However, in another app, "( . . . ) the technical level fails to immerse" (1943 Berlin Blitz). Hence, immersion and audio-visual technology are often mentioned together.

Next, the storytelling or script underlying the immersion (3), as well as the background information given (4), both generated user comments: Most of the positive comments referred to the VR experience Is Anna OK?, since "they connected [ . . . ] two stories together", which "gives you two different perspectives"; moreover, the experience "is based on a true story where the script is written from their memories". It has been summarized as "a great example of good storytelling in VR". The comments about storytelling in another app (Home—A VR Spacewalk), however, were all negative; here, "the design decision to trade a real astronaut experience for a sci-fi/horror film ending where it is impossible to make it back alive is just insulting to the end users"; similarly, "in the ISS version you have complete freedom of movement whereas here you're pulling yourself along by heavily predefined points". Hence, the perception of the script and the content also influenced the perceived immersion. Lastly, the technical implementation that causes immersion (5) was also mentioned by the users, usually in a rather critical tone, i.e., "the technical level fails to immerse" (1943 Berlin Blitz) or "what would bring more emersive [immersion] to this experience is ( . . . ) real instrumentation, and the animations were a bit poor ( . . . ) (1943 Berlin Blitz).

### 4.2. Emotion: "This VR Experience Made Me Cry"

The main category emotion also generated four inductive subcategories: Firstly, users frequently and positively commented on the general emotional character of the app/story (5), writing that "I don't know why I got so emotional, but I was hit hard" (1943 Berlin Blitz), the app "made me teary" (Home), or "this is the first VR experience that I have tried that made me cry. It was beautiful, insightful and above all meaningful to me" (Notes on Blindness). They also mentioned an "empathy building experience" (We Wait).

Secondly, within the perception of audio-visual effects and sensuality (6), users positively emphasized "the reporter's real commentary" (1943 Berlin Blitz), or "the recordings which were very eloquent and powerful" (Notes on Blindness), but also criticized low-quality visuals: "there is a difference between a minimalistic art style and one which actually inhibits the viewer's ability to empathize with those represented. The faces of the characters especially were off putting" (We Wait). Hence, the degree of felt emotion also stands in relation to aspects of technology, which confirms the earlier-mentioned interplay of the two.

Concerning the storytelling or script that causes emotion (7), nearly all comments highlighted positive aspects, such as "great VR experience and emotional storytelling in VR" (1943 Berlin Blitz), or "the future of storytelling in VR. ( . . . ) What an emotional but also powerful experience" (Is Anna OK?). Similar to the connection between immersion and storytelling, the users' demands for storytelling and background information (8) also seem to be very high when it comes to emotion. If the background information is accurate and functions well, the whole VR experience benefits: "As a life-long student of history, I have never witnessed the stench and naked fear of actual war as experienced in this immersive heart-pounding fear-gripping display" (1943 Berlin Blitz). Lack of content, however, lowers the overall experience, and thus also the emotional one: "They cut off the ending and the beginning, so you're left with no idea why they're invading Europe" (We Wait).

### 4.3. Usability: "The Overall Quality Was Very Poor"

In contrast to the rather positive tone of the comments concerning immersion and emotion, the comments regarding usability and utility were mostly of a negative, critical tone, and positive comments only referred to the absence of motion sickness (1943 Berlin Blitz). Here, again, the aspects of background information and factualness were evaluated as highly important for the overall experience of usability. Firstly, users critically referred to the perception of audio-visual effects (9), stating that "the graphics ( . . . ) are not the best" (1943 Berlin Blitz), "the sound of flak is disappointing" (1943 Berlin Blitz), or "the

camera quality is so low that anything beyond 4 feet away is too blurry to bother watching" (CNN VR).

Secondly, as already mentioned, the fact that users "didn't get motion sickness" (10) (Berlin Blitz) appeared as one category within the realms of usability. The navigation and control (11) among these apps, however, were addressed most frequently and negatively, for example as "terrible locomotion" (Home), "no control over experience" (Bear Island), or "this game's mechanics are terrible" (Home). In line with that, the interactivity (12) of the apps, as a third dimension within the usability category, was also judged critically, as being a "10-min non-interactive presentation" (Buzz Aldrin), or "a vision without any technical details and without any interaction" (Buzz Aldrin). Through these comments, it becomes obvious that the users' expectations of the VR experience do not seem to have been fulfilled.

The aspect of background information (13) was also connected to usability, and received at least as many negative comments as the navigation and control. Commentators wrote that "it's just silly ( . . . ), only has ideas and no tech" (Buzz Aldrin), "there are a few scientific flaws in the presentation" (Buzz Aldrin), or "it would be nice if you could explore other objects in the library" (Sky VR). Some users wished for more content in order for the story to be even more useful to them (VR; We Wait), but at the same time, they also appreciated the content: "We need to see more VR content like this" (Sky VR), "the content is great" (Sky VR). Last but not least, language (14) was referred to in terms of a lack of subtitles (1943 Berlin Blitz, Home, Sky VR) or translations (Berlin Blitz, Easter Rising, Home, Sky VR). Hence, the general usability of VR apps would benefit from translations and subtitles, and great potential seems to lie in the displayed content.

### 4.4. Utility: "An Obsolete Technology"

Aspects of utility were mentioned in 14 out of the 15 analysed apps; thus, these aspects seemed relevant in a broad variety of applications. Nearly all of them had a negative tone, especially with regard to technical problems and implementation. Concerning the technological requirements for installation (15), users mentioned general problems such as "can't even start it playing" (Cat Flight), or "can't even install it" (Home), and also referred to the required software—such as Windows 10 (Sky VR)—the required internet connection (Sky VR), or the download capacities (We Wait). Positive comments referred to the size of the app, such as "runs fine on 24 GB of RAM" (*NYT*).

When it came to the general technical factors in terms of utility (16), several users across a majority of the analysed apps had problems even making the application work (Cat Flight, Bear Island, Buzz Adrenalin, CNN VR, Home, Notes on Blindness, Oogie, Sky VR), leaving comments such as "app is broken!" (Sky VR), "flat out doesn't work" (CNN VR), "it lacks any depth and has all kinds of technical and other issues and just feels very unpolished" (Buzz Aldrin), or "useless app" (Sky VR).

Finally, these flaws also had an impact on the technical implementation of audio-visual effects (17), which were also commented on by the users. Here, the main criticism across the apps lay in the lack of 3D (Bear Island, Cat Flight, CNN VR, Cut-Off). One user even commented "this is actually the worst 3D experience I had in Oculus Rift" (Cat Flight), and another cynically asked "How can you make 3D one dimensional? By limiting UI's [user interfaces] to this pathetic pseudo-VR implementation of an obsolete technology" (CNN VR).

The technical implementation of audio-visual effects, however, was also commented on in a more positive way, with users stating that "VR is getting better and better" (Is Anna OK?), and that an application was "very, very technically sound, no hitches of even the slightest amount" (Is Anna OK?). Further comments appreciating the utility of the audio-visual component were made, such as: "Technically, this really shows what is possible with photogrammetry and high-resolution scanning in VR. This is a truly well-crafted piece of VR entertainment" (Sky VR).

*4.5. Discourse: "Ill-Informed Conspirational Nonsense"*

The category of discourse was the only main category that emerged inductively on the basis of the analysed comments, since aspects of political discourse and propaganda (18) and fake news (19) appeared within the comments. While the four deductive main categories appeared throughout all apps, the discourse category only appeared within three apps, which will be explicitly mentioned here: the app 1943 Berlin Blitz, about a BBC reporter who boards a plane in 1943 on an allied bombing raid over Berlin; the app CNN VR as a newsroom of the future; and the app We Wait, circling around the refugee crisis, all generated political comments and discussions.

Concerning political discourse and propaganda (19), 1943 Berlin Blitz was referred to as "propaganda" and "propaganda esque". Within We Wait, a discussion about liberalism, right-wing extremism, and xenophobia emerged. Users accused the app of being "politically motivated, worse than watching CNN" and being "political crap", and seemed to criticize those responsible for the app with comments such as "liberals are now on VR". Others criticized the right-wing tendencies and comments of users, even connecting the dots back to the potential for empathy: "They're probably male, young and not well-educated persons [who] are not able to be empathetic—I guess they never learned it. ( . . . ) They are so full of hate that they even have to shout out on a review board". Another user wrote "you people will look back on your comments with regret".

The political discourse on CNN VR mostly referred to the news outlet itself, with users calling CNN the "Clinton News Network". One user commented "this is not a genuine news outlet, this is a proper gander [propaganda] outlet full of nothing but hysterical nonsense ( . . . ). They should be shut down!". In that regard, the second discourse dimension—fake news (20)—emerged, with users stating "Fake news CNN. ( . . . ) The communist news network propaganda in VR glory", or "fake news in VR. Neat, can get all my communist propaganda beamed directly into my brain now. We truly live in the future". While these comments partly have a cynical undertone, they show that VR clearly offers a platform for political discourse. This adds an important layer to the discussion about VR in journalism, showing that it is not only the technological features of the VR experience and the factualness of the content that shape a user's experience and his or her degree of immersion and generated emotion, but also the user's personal disposition towards certain topics. On the one hand, this seemingly very individual layer cannot easily be influenced by journalistic reporting, and surely also applies to other forms and formats of journalism; on the other hand, VR journalism in particular could have the potential to distribute different points of view and to enhance societal discourse.

## 5. Discussion and Summary: Empathy Machine or Digital Spectacle?

Taking a look at the inductive categories and the descriptive results of the analysis of user comments, patterns and main trends are visible, which answer the guiding research question of how users of journalistic VR apps evaluate these apps in terms of immersion, emotion, usability, and utility.

First of all, it is clear that throughout the whole analysis, users positively commented on aspects of immersion and emotion, and negatively on aspects of usability and utility. This represents an initial starting point for a subsequent quantitative analysis of user comments:

**Hypothesis 1 (H1).** *The majority of users commenting on journalistic VR apps refer to immersion and emotion in a positive tone.*

**Hypothesis 2 (H2).** *The majority of users commenting on journalistic VR apps refer to usability and utility in a negative tone.*

If the majority of users commenting on journalistic VR apps refer to immersion and emotion in a positive tone (H1), it would be of interest to analyse to what degree immersion and emotion are associated with one another:

**Hypothesis 3 (H3).** *Positive perceptions of immersion in journalistic VR apps are associated with positive perceptions of emotion.*

This may indicate where journalistic media organizations that want to advance in VR production should focus their attention. While the narratives, storylines, and content seem to be in place (and should, of course, continue to be treated with equal amounts of attention and care), a stronger focus on the technical implementation seems relevant now. Hence, rather than a fear of "placing too strong an emphasis on the technology" (Hand 1994), there should probably be a concerted effort to place a strong emphasis on it in order to enhance the levels of utility and usability, from which levels of emotion and immersion will also ultimately benefit.

In the same vein, however, it would be worth analysing whether emotion and immersion can exist despite flaws in technology. This would be especially interesting for low-budget journalistic outlets that might not be able to keep up with the latest VR technology, but still want to immerse their users into their carefully crafted content and provide an emotional experience. Hence, further studies could investigate whether users who comment on the usability of a VR app in a negative way might at the same time appreciate the emotional or immersive experience in a positive way:

**Hypothesis 4 (H4).** *Negative perceptions of usability of journalistic VR apps exist simultaneously with positive perceptions of immersion and emotion.*

Furthermore, the occurrence of the fifth inductive discourse category showed an inverse, related danger—or even opportunity—for VR in journalism when it comes to elevated degrees of emotion: For controversial topics, VR apps can also be (mis)used as a playground for misanthropic behaviour. Although this might offer the chance for societal discourse provided by journalism, journalistic media organizations need to keep the ethical dimensions of their projects in mind. This seems to represent a thin line to walk: while on the one hand, VR journalism could enhance political discourse about controversial topics, and even offer the potential for two opposing sides to understand one another better, the risk of verbal escalation and destructive communication, on the other hand, seems very high. Possible touchpoints could be discussion forums, moderated discourses, or further tools of community management, as offered by the respective journalistic organizations. Since the discourse category only appeared within three apps, the following hypothesis only stands on these grounds; however, it was integrated for its seemingly high relevance for journalistic and political discourse, and future analyses could and should investigate this further:

**Hypothesis 5 (H5).** *VR apps of journalistic media outlets can stimulate political discourse.*

According to a (probably invented) anecdote, the audience at the premiere of the early cinema film *L'arrivée d'un train en gare de La Ciotat* (1895) is said to have fled the auditorium in terror when a steam locomotive arrived in a station during the 50-s recording (Loiperdinger 2004). This was made possible by the immediacy effect of the new technology, which charged the film's actually manageable plot with unexpected force. Scholars argue that humans have always succeeded in integrating new forms of reality (simulation) into their lives, as was the case with the train movie (Morat 1998, p. 43). When applied to VR, however, it can be said that media organizations sometimes seem to believe that they are dealing with an audience comparable to that of the early film industry. There is no other explanation for the fact that, despite initially promising approaches to establishing new forms of presentation, they are often not succeeding in inspiring a broad audience for their

projects. This is reflected not least in the simple observation that all of the apps analysed together achieve just a third as many ratings as the particularly popular VR game Beat Saber on the same platform.

Digital journalism needs to find its strengths and signature formats in VR. A crucial point is, and will ever be, the technological development, which of course waits for no one and requires constant adaptations and progress. Courageous, adventurous, and technically advanced journalistic media organizations could make use of this momentum in not yet fully developed VR app solutions and become leaders in their field. The educational and informational possibilities that this format offers should not be neglected, and should be focused on. Current trends within the realms of artificial intelligence or deepfakes could be made use of in order to broaden the horizons of users—both virtually, and in reality.

## 6. Limitations

This qualitative, explorative paper does not claim completeness or representativeness. It does, however, present an examination of the tendencies in the user comments on the journalistic VR apps under analysis. As is inherent in the nature of user comments, posting a comment is a purely self-selective phenomenon, and it cannot claim external validity for all users of the Oculus Store. Instead, it reflects the subjective opinions of those who wanted to articulate their praise or criticism in a particular way. Accordingly, despite the relatively high number of comments analysed, it did not seem to make sense to show statistical frequencies, especially since the present study had an explorative character. Some apps were reviewed more often than others, and this aspect, paired with the findings within the individual apps, is the starting point for a quantitative analysis of the user comments. For this, the explorative study was necessary in order to generate a first set of inductive categories and hypotheses, with the help of which the apps can now be further analysed in a quantitative manner. Concerning the content analysis, the categorization of positive and negative tone in particular remains a rather subjective undertaking, including the detection of sarcasm or cynicism within the comments. Although the two coders ran a pre-test and received coder training, this aspect should not be neglected. The three developed and deducted hypotheses, however, can now be used as a link to a further quantitative study, in which the individual apps with their statistics can be analysed in detail.

**Author Contributions:** Conceptualization, A.G.; methodology, A.L. and F.V.; software, A.L. and F.V.; validation, A.G., C.W. and R.P.; formal analysis, R.P.; investigation, R.P., A.L. and F.V.; resources, A.G.; data curation, A.L. and F.V.; writing—original draft preparation, R.P.; writing—review and editing, A.G. and C.W.; visualization, A.G. and R.P.; supervision, A.G. and C.W.; project administration, A.G. and C.W. All authors have read and agreed to the published version of the manuscript.

**Funding:** This research received no external funding.

**Institutional Review Board Statement:** Not applicable.

**Informed Consent Statement:** Not applicable.

**Conflicts of Interest:** The authors declare no conflict of interest.

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
