# Peer review of "An Immersive Journey through Flawed Technology: Users’ Perceptions of VR in Journalism"

_journalmedia, doi:10.3390/journalmedia2030027_

Round 1

Reviewer 1 Report

The manuscript aims at exploring the user experience of immersive VR journalism by analyzing N=770 user comments left for 15 apps produced by news media organizations and offered in the Oculus Rift store.

While the topic is timely and relevant, the manuscript, unfortunately, comes with several problems:

  • “VR technology captures the urge of humankind to explore the space between reality and virtuality.” -> I could not understand this apodictic and opaque sentence that comes without any reference. Avoid speculation about all “humankind” in an empirical paper
  • For definition of VR use a more authoritative source form the international VR field
  • Same for other key concepts (immersive journalism, immersion, presence): Use definitions from original authors
  • “Digital journalism” is used without any definition
  • “For instance, users of VR applications objectively have no reason to experience motion sick-ness, since they are standing safely in a room, wearing VR glasses – but because the human body perceives the virtual reality as so real, they often report feeling nausea or motion sickness.and virtuality..” -> inaccurate claim, current research shows that the causes of cybersickness are definitely unrelated to the perception of realness of the scene
  • The presentation of state of research with its sub-chapters does not match the main deductive categories used later in the analysis. Make sure the theory part matches those four dimensions (table 2) – otherwise the readers are confused
  • “the lion’s share” – use factual language instead of exaggerations
  • One MAJOR problem of the whole paper is that authors talk about “journalistic VR apps” but do not clearly define them. That apps were produced by news media organizations does not imply that the apps are “journalistic apps” per se. In their self-descriptions the selected apps are often introduced as “games” or “learning games”. So it is highly questionable if their content truly meets standards of journalism. It is alarming that the authors do not provide any definition and do not address this issue at all. I highly doubt that the selected apps are “journalistic” and, hence, that the data speak to the experience of “immersive journalism” please clarify and make sure not to over-interpret the data
  • I also question if the different apps are comparable in term of content and VR experience
  • Selected apps need to be introduced with some description of their content and immersive part. Without this background information readers cannot make sense of quotes that refer so selected apps.
  • The methodology is non-transparent. It remains unclear who coded what. No documentation of coding material. No mention of qualitiative data analysis software. No mention of any measures of quality control (e.g. reliability measures).
  • Deductive categories are introduced in Table 2 but their sources are missing
  • A section on research ethics is missing
  • A sample description of the comments (publication dates, length etc.) is missing
  • “despite the high number of comments analyzed” -> I don’t agree with the high number, for several apps less than 20 comments were covered and 770 short comments is not a large sample
  • I don’t understand the rationale and the merit of the hypotheses building in the discussion section. Are you generalizing from 15 mostly outdated apps in the Oculus Rift store to the whole future population of “journalistic VR apps” – does not make any sense to me.
  • Paper does not address the problem that most selected apps several years old.
  • Overall I was disappointed that the discussion does not speak to the VR experience specifically in the context of journalistic messages – the issues with usability are independent from content and affect fictional content just the same. So, I think it’s hard for readers to understand what we have learned about VR in Journalism.

Reviewer 2 Report

This article addresses a topic of interest, such as users' perceptions of virtual reality in journalism, with a critical and realistic perspective on the current state of immersive journalism.

I suggest introducing in the title that the analysis of perception is done through the analysis of comments in the Oculus Store. Assessing user perception based solely on comments is a very limited view of the subject. A review of the type of feedback typically found on content platforms and app stores shows that in many cases the comments area is used to send complaints. However, the authors recognize this limitation and others, so the article meets the expectations of an exploratory study.

The state of the art includes the main milestones in the evolution of immersive technologies. However, it could be updated with some contributions from the book:

Uskali, T., Gynnild, A., Jones, S., & Sirkkunen, E. (2021). Immersive Journalism as Storytelling: Ethics, Production, and Design. Routledge.

The anlysis sample is justified and relevant, within the Oculus Store offer.

Regarding the results, the structure by category allows a panoramic view. However, I miss knowing to what extent (quantitatively) the comments of the cases analyzed were of one type or another, both positive and negative; as well as an indication of the most positively highlighted cases (some of them with a significant number of comments).

Another aspect that should be considered is that in the sample there are independent VR pieces and repository-type applications (such as CNN VR or Sky VR), as defined by Vázquez-Herrero and López-García in "Immersive journalism through mobile devices: How virtual reality apps are changing news consumption". This point is relevant because the analysis must consider that in an app like Notes on Blindness users will only comment on that story, but in CNN VR they can find several VR pieces.

The hypothesis proposal the authors leave at the end of the article is interesting and justified, although it refers only to the comments left by users (from my point of view, it is a limited view of reception).

The big question that remains to be answered is why it is a flawed technology and they could trace some clues in the conclusions.

Mistakes they need to correct:

"De la Pena" (p. 3).

"Sànchez Laws" (p. 4)

Round 2

Reviewer 1 Report

I think the revision has improved the manuscript.

The information on research ethics provided in the answer letter should be integrated in the manuscript. Also, authors should explain with references to the online research ethics papers why they think they can use the customer reviews without the commenters' consent. 

I still don't understand why the authors cannot share their coding material in an e-appendix or on osf.io. I think this weakens the value of the manuscript. We not join the open science movement? 

Why has no hypothesis been suggested based on the category: Discourse? I find this inconsistent.

Author Response

Hello,

and thank you for your constructive new comments! We integrated the section on research ethics into the manuscripts and also elaborated on it further, including references.

The coding material consists of several documents and is not formatted or edited as of now. We are, of course, in favour of the open science movement and are more than happy to share the data with anyone who wants to know more about it. The corresponding author is always happy to reply to such requests. 

Thank you for pointing out the inconsistency regarding hypothesis 5! Initially, we did not include it since it was only based on the comments made in three apps, as stated. We agree, however, that it seems inconsistent and added the hypothesis, explaining the background of it. 

Thank you again and best regards!
